# Design and Mechanical Characterization Using Digital Image Correlation of Soft Tissue-Mimicking Polymers

**DOI:** 10.3390/polym14132639

**Published:** 2022-06-28

**Authors:** Oliver Grimaldo Ruiz, Mariana Rodriguez Reinoso, Elena Ingrassia, Federico Vecchio, Filippo Maniero, Vito Burgio, Marco Civera, Ido Bitan, Giuseppe Lacidogna, Cecilia Surace

**Affiliations:** 1Department of Structural, Geotechnical and Building Engineering (DISEG), Politecnico di Torino, Corso Duca Degli Abruzzi 24. P. C., 10129 Turin, Italy; oliver.ruiz0809@gmail.com (O.G.R.); mariana.rodriguez@polito.it (M.R.R.); s267645@studenti.polito.it (E.I.); federico.vecchio@polito.it (F.V.); s257565@studenti.polito.it (F.M.); vito.burgio@polito.it (V.B.); giuseppe.lacidogna@polito.it (G.L.); cecilia.surace@polito.it (C.S.); 2Laboratory of Bio-Inspired Nanomechanics “Giuseppe Maria Pugno”, Politecnico di Torino, Corso Duca Degli Abruzzi 24. P. C., 10129 Turin, Italy; 3Stratasys Headquarters, 1 Holtzman St. Science Park, Rehovot P.O. Box 2496, Israel; ido.bitan@stratasys.com

**Keywords:** additive manufacturing, PolyJet technology, polymer, Shore A hardness, mechanical properties and standards, finite element analysis, uni-axial tensile test, digital image correlation, tendons and ligaments

## Abstract

Present and future anatomical models for biomedical applications will need bio-mimicking three-dimensional (3D)-printed tissues. These would enable, for example, the evaluation of the quality-performance of novel devices at an intermediate step between ex-vivo and in-vivo trials. Nowadays, PolyJet technology produces anatomical models with varying levels of realism and fidelity to replicate organic tissues. These include anatomical presets set with combinations of multiple materials, transitions, and colors that vary in hardness, flexibility, and density. This study aims to mechanically characterize multi-material specimens designed and fabricated to mimic various bio-inspired hierarchical structures targeted to mimic tendons and ligaments. A Stratasys^®^ J750™ 3D Printer was used, combining the Agilus30™ material at different hardness levels in the bio-mimicking configurations. Then, the mechanical properties of these different options were tested to evaluate their behavior under uni-axial tensile tests. Digital Image Correlation (DIC) was used to accurately quantify the specimens’ large strains in a non-contact fashion. A difference in the mechanical properties according to pattern type, proposed hardness combinations, and matrix-to-fiber ratio were evidenced. The specimens V, J1, A1, and C were selected as the best for every type of pattern. Specimens V were chosen as the leading combination since they exhibited the best balance of mechanical properties with the higher values of Modulus of elasticity (2.21 ± 0.17 MPa), maximum strain (1.86 ± 0.05 mm/mm), and tensile strength at break (2.11 ± 0.13 MPa). The approach demonstrates the versatility of PolyJet technology that enables core materials to be tailored based on specific needs. These findings will allow the development of more accurate and realistic computational and 3D printed soft tissue anatomical solutions mimicking something much closer to real tissues.

## 1. Introduction

The adoption of additive manufacturing (AM) in healthcare has significantly changed the therapeutic paradigm of personalized medicine, thus enabling the execution of complex treatments and surgical procedures by clinicians and hospitals. In the present day, major solutions in the medical field are divided into two categories, (1) applications of patient-specific anatomical models and (2) the manufacturing of custom-made medical devices [1,2,3,4,5]. For anatomical models, trends include but are not limited to: diagnosis, education, research, and development, training of healthcare providers, custom-made medical device development and testing, and pre-operative procedure planning [6,7,8,9,10]. In addition to the above, the more realistic anatomical models in combination with modern AM processes have the potential to experiment and evaluate the quality-performance of novel implantable devices at an intermediate step between ex-vivo and in-vivo trials or to plan and perform precise demanding surgical procedures in non-risk settings. Reducing the succeeding use of cadaveric and animal models [11,12,13,14,15,16]. In this regard, anatomical modeling aims to overcome two main challenges. The first is to improve the hardware of three-dimensional (3D) printers, particularly due to limitations associated with dimensional and geometrical accuracy at the micro-nano level, long printing times, and high costs among others. The second regards, the development of new 3D-printable tailored materials that match the biomechanical properties of real tissues [17,18,19,20,21,22], also considering appropriate curing and post-processing [23]. The demand for materials that react like human tissues might be addressed by considering new methods, one of them is based on a bio-inspired approach by properly mimicking the unique, complex characteristics and behavior of the biological systems. Major efforts in this field are aimed at creating anatomical models with more use of multi-color and multi-material structures. However, beyond all current advances in AM hardware in this field, these are limited by the lack of suitable printing materials with printability and desired mechanical properties. Even as bioprinting evolves and becomes more accessible with advanced therapy products through producing bioreactors and tissue constructs, its implicit ethical difficulties and likely high costs will allow tailored and composite materials to remain relevant [24,25,26,27,28,29,30].

Among the various AM technologies, the photopolymer jetting processes are the most suitable options for multi-material and multi-color anatomical modeling solutions. This type of technology shows several advantages in anatomical modeling such as (1) the capability to manufacture a single model in multiple materials, colors, textures, blended transitions, and transparencies (e.g., a knee joint including bone, vessels, ligaments); (2) it enables obtaining high-performance composite materials, combining hard and soft photopolymers in a single process. (e.g., a bone printed from rigid and elastic materials that mimic compact-spongy tissues and the transition between both); and (3) the calibration of the printing materials enabling variations of mechanical properties (e.g., healthy and pathological heart tissue printed with the same material but with different levels of stiffness and textures) [12,31,32,33,34,35,36,37]. In this context, the Stratasys J750™ digital anatomy 3D printer (DAP) (Stratasys, Eden Prairie, MN, USA) includes in addition to the PolyJet™ materials (flexible, stiff, soft, and hard materials) a suite of preset digital anatomy materials, designed to mimic the human anatomy. The 3D printing parameters can be set to emulate a similar haptic feel, responsiveness, and biomechanical performance of real tissues. This allows as well for the precise definition of complex bio-inspired shapes [38]. The entire 3D printing settings can be controlled by GrabCAD print™ software voxel-based system, which enables modifying physiological characteristics of preset materials [31,39,40,41]. The GelMatrix™, TissueMatrix™, and BoneMatrix™ are the trade names of the three anatomical digital materials available in the printer, which are used to suit the requirements and specifications of anatomical applications. Each material has different mechanical characteristics, TissueMatrix™ is appropriate for reproducing the appearance and structure of heart tissue because it is the softer material available. BoneMatrix™ is a stiff material with shape memory, used to produce musculoskeletal models. Finally, GelMatrix™ is a gel-like support material used to print small and complex vascular models. Each digital material in the software is configured using PolyJet™ material combinations that can vary in hardness measurable in terms of the Shore A scale, elasticity, and density. The multi-material jetting technology enables the manufacturing of realistic, functional, and innovative structures integrating a hard-soft phase and tailored properties, without compromising the complexity of the geometric design [41,42,43,44,45]. All these features are required to replicate the mechanical behavior of the target biological systems. Indeed, to achieve their intended needs, in nature, the biological systems combine hard and soft phases in their hierarchical structures. The natural patterns are often highly complex to perform an identical reproduction. Because of this, the man-made bio-inspired approach often has limitations and is made up of a limited number of constituent elements. The biological systems with hierarchical architectures offer a unique inspiration for designing and manufacturing composite materials that can demonstrate outstanding mechanical performance and the needed level of fidelity. In this context, the photopolymer jetting AM process has demonstrated great potential in assembling these structures aimed at various applications in the medical field where adequate stiffness and flexibility are required simultaneously [46,47,48,49,50,51,52]. In particular, the bio-inspired approach in combination with PolyJet technology can be applied in the search for new composite materials targeting complex specialized tissues. However, despite all these advances in the surgical field, it is widely known that studies about the repair of major joints remain challenging. One of the major limitations of present preoperative and interoperative models is the lack of printable materials suitable to emulate specialized soft tissues, such as tendons and ligaments. These are responsible for joint stability and play a key role in the transmission of loads thus allowing the movement across the joint range of motion (ROM) [53,54,55,56]. Their hierarchical structure enables supporting high rates of stress and strain without suffering irreversible damage [57]. Most of the investigations are focused on the printing of 3D scaffolds for tendon and ligament repair [58,59,60]. The manufacturing of a multi-material model that mimics the mechanical behavior of those soft specialized tissues using tailored polymers available in the Stratasys J750™ DAP remains almost unexplored.

This study aims to evaluate different bio-inspired patterns that mimic hierarchical structures like those present in tendons and ligaments combining 3D modeling, finite element analysis (FEA), multi-material additive manufacturing, and mechanical characterization. The present approach is structured as follows: First, the design of specimens with the embedded patterns (described as patterned specimens) using computer-aided design (CAD) software considering natural fractal structures like the ones of the tendon itself, bamboo, and collagen. Before the mechanical testing of the dog-bone specimens, FEA was performed to ensure the most suitable strategy to place the bio-inspired pattern inside the test pieces. Second, the Stratasys J750™ DAP was used to produce the patterned specimens, combining different Agilus30™ durometers between the matrix and the patterns. The mechanical characterization was performed by two methods, the uni-axial tensile test (UT) and digital image correlation (DIC). In particular, DIC was used for the measurement of normal and longitudinal strain fields. this non-contact optical technique arises to be the most appropriate to evaluate accurately the large volumetric strains (>50%) of materials that exhibit anisotropy such as biological tissues [61,62,63,64] or composite materials (e.g., [65]).

## 2. Materials and Methods

### 2.1. Three-Dimensional Modeling and Printing

The present study was performed in the Bio-Inspired Nanomechanics laboratory of the Politecnico di Torino. The dog-bone specimens were printed at the Stratasys headquarters (Rehovot, Israel) using the Stratasys J750™ DAP, a multi-color and multi-material 3D printer. The printing mode selected was the High Mix mode (the print-head set-up includes seven materials loaded plus support, and 0.027 mm layer thickness), and the orientation of the specimens on the building tray was along the *z*-axis following the building direction. The entire dog-bone specimens and biomimetic patterns tested were designed using Rhinoceros 3D^®^ CAD software (Robert McNeel & Associates, Washington, DC, USA) following dimensions reported in the ASTM Standard D638 Type IV tensile designation [62]. The specimen shape and dimensions described are reported in Figure 1.

The PolyJet™ material Agilus30™ (FLX935), was selected as the most suitable candidate for the proposed polymer combinations [39,66]. This is a rubber-like photopolymer capable of withstanding repeated flexing and characterized by Shore A hardness ranging from 30–95 and tensile strength at break between 2.4–3.1 MPa with a maximum strain at break between 2.2–2.70 (mm/mm) according to its datasheet. Initially, three different Shore A hardness values were assigned to the Agilus30™ material specimens using the GrabCAD™ printer software. In particular, the Shore A hardness values of A50, A60, and A70 were chosen. In addition to this, TissueMatrix™ material was also included for comparison purposes (Specimens ID: T1 = Agilus30™ Shore hardness A50, U1 = Agilus30™ Shore hardness A60, V1 = Agilus30™ Shore hardness A70, and W1 for TissueMatrix™). To prevent failure on the region of the clamping specimen during the test, due to the pressure applied by the tensile machine grips, all specimens’ attachments (ends) were printed in VeroWhite™ (RGD835) a rigid material as shown in Figure 2. An overlapping of 1 mm between this material and the Agilus30/TissueMatrix™ was designed to prevent high-stress values due to discontinuity at the interface. Water cleaning was made 72 h before the tensile test, no additional chemicals were used in this process [12,16].

### 2.2. Finite Element Analysis and Evaluation of Pattern Position Inside the Specimens

A Finite Element Analysis (FEA) was performed to evaluate the best position of the pattern inside the specimens for a mechanical tensile test. Applying the same soft tissue design approach as [40]. The cylindrical linear fiber patterns with a diameter of 0.5 mm were embedded in a matrix with the shape of the specimens. We laid two different arrangements of the fibers among the longitudinal direction of the specimen: Configuration (I) fibers uniformly distributed in the whole volume (Figure 3A) and Configuration (II) fibers positioned only in the narrow cross-section of the specimen (Figure 3B). All assembled specimens were exported as standard for the exchange of product model (STEP) files from Rhinoceros 3D, to ANSYS 2020 R2^®^ (Swanson Analysis Systems, Inc., Houston, PA, USA) Computer-aided engineering (CAE) system. The simulations were performed by setting two Shore A hardness values A50 and A60 for Agilus30™ material for fibers and matrix, respectively. The meshing process was done using a tetrahedral formulation with a type of unit selected of 10-node elements (TED10) for all components (fibers and matrix). This type of element was selected because it allowed the reproduction of properly curved surfaces with curved edges, owing to each edge of the volume element having mid-side nodes. For the first configuration, 771,235 elements for the fibers and 264,742 for the matrix were set to fit geometry shapes. For the second configuration, 145,941 and 132,824 elements were set, respectively.

Some hypotheses were considered to simplify the complexity of the simulation. The Agilus30™ material was considered a homogeneous, isotropic linear elastic material. The tensile material properties, such as modulus of elasticity, tensile strength at break, and maximum strain at break were set in Ansys static structural environment using the experimental values obtained in Table A1 (Appendix A) reported in the results section for base PolyJet materials. A density (1.15 g/cm^3^), and a Poisson’s ratio (0.34) were set according to the values reported in the literature [66,67]. To simulate a uni-axial tensile test, fixed support was set on the base of the specimen and a displacement of 50 mm was applied in 10 steps of one second each. The contact regions between fibers and matrix were selected as bonded, to emulate the real test.

### 2.3. Tensile Test Framework

The mechanical tensile test of the specimens was performed using the MTS Insight^®^ Electromechanical Uni-axial Testing System with a 100 N load cell and a sensitivity of 2.164 mV/V. For this study, the dog-bone specimens were subjected to uni-axial tensile stress along their longitudinal axis (*z*-axis). The test was conducted on three type IV dog-bone specimens (n=3) for each set of patterned specimen combinations and base PolyJet polymers evaluated in this study. A strain rate of 50 mm/min, a sampling rate of 60 Hz, and a grips separation of 65 mm were set as regulated by the standard. The dimensions (thickness, length, width) of each specimen were measured with a caliper, the parameters were entered into the software TestWorks^®^ Software-MTS Systems Corporation. The mechanical properties: tensile strength at break (MPa), maximum strain at break (mm/mm), and modulus of elasticity (MPa) were calculated following the relations outlined in the standard D638 Type IV Tensile Designation. The Pearson’s coefficient (R^2^) was computed by making a linear regression on the stress–strain curves of the specimens, using a linear function on Microsoft Excel^®^ (Microsoft Corporation). The coefficient of variation (CV%) was calculated as the ratio of the standard (std) deviation to the mean. Figure 4 shows an experimental mechanical testing setup for dog-bone specimens. It shows the deformation of the specimen in the initial elastic deformation stage until failure.

### 2.4. The Bio-Inspired Approach of Soft Tissue-Mimicking Specimens

Following a bio-inspired approach, four biological structures were foreseen to mimic the mechanical properties of tendon and ligament tissues. The designed patterns were embedded in a matrix, both with different Shore A hardness levels aimed to identify the most suitable combination. Shore A values A50, A60, and A70 were considered; higher values than A70 were not included due to the high stiffness. Based on the mechanical principle of tendons and ligaments enabling the transmission of physiological loads between muscles and bones [68], a Tendon-Like (TL) pattern that mimics the hierarchical structure of the tendon on a micro-and mesoscopic scale was proposed. Thus, considering one fiber inside the other and so on, as shown in Figure 5, the structure was designed to recreate 14 tendon fascicle structures in the cross-sectional area (CSA) of the specimens, each fascicle containing three sub-fascicles and each of them containing one fiber. In particular, the inner layer refers to the single fiber (1–20 mm), while the middle and outer layers refer to the sub-fascicle (20–100 mm) and tendon fascicle (100–500 mm), respectively [69]. Each level or layer was assigned a different Shore A hardness value as shown in Table 1.

Our second approach was titled Tendon-Mimic (TM) pattern; it was designed by mimicking the non-uniform fascicle dimension of equine superficial digital flexor tendon (SDFT), examined using an SMZ18 Research Stereo Microscope. The fascicles were segmented on Rhinoceros 3D software to calculate the CSA of each fascicle’s silhouette. Subsequently, three fascicles were chosen, which showed a more regular shape and had a CSA close to the mean. By modifying the dimension of the three fascicles, some variants were created, considering the percentage of area that was occupied by the fibers on the narrow cross-section of the specimen. Thus, a fibers percentage of Variant A (50%), Variant B (60%), and Variant C (33%) were considered as shown in Figure 6. The design specifications according to the Shore A hardness values assigned are given in Table 2.

The third approach followed a natural arrangement, which was titled Bamboo-Like (BL) pattern. It mimics the concentric layer arrangement of the bamboo fibers. The selection of the bamboo architecture was based on the exceptional structural efficiency in terms of mechanical performance per unit weight of bamboo. This property derives from the optimization of their cellular microstructure and tubular macroscopic shape [70]. The BL hierarchical structure was designed by recreating 13 fibers of bamboo in the CSA of the specimens, each of them containing three concentric layers as shown in Figure 7. The design specifications according to the Shore A harness are given in Table 3. The bamboo cortex fibers are hollow tubes composed of several concentric layers and each layer is reinforced with helically wound microfibrils (or protofibrils) [71].

Our last pattern, titled Helix Bamboo-Like (HB) pattern, considers a single bamboo fiber that is composed of three nested layers. Considering the bamboo microstructure, the two outermost layers contain fibers with a double helix configuration. In particular, the helix pitch selected was 2.5 and 3.5 for the external and the middle layer, respectively, while in the innermost layer there is a single cylindrical fiber as shown in Figure 8. As regards, HB specimens there were considered the same values of Shore A hardness for the specimen matrix and fibers, while the matrix of each layer was set with different Shore A values as shown in Table 4.

### 2.5. Digital Image Correlation (DIC) Setup

For examining the influence on the mechanical behavior of the single patterned specimen, in addition to the tensile test, Digital Image Correlation (DIC) was used to closely track the volumetric deformations. The DIC system implemented was the VIC-EDU^®^ 8 System (Solutions—Correlated Solutions, Inc., Columbia, SC, USA), including two stereo high-speed cameras FLIR Blackfly^®^ U3-23S6M-C, with a progressive scan “CMOS 1/1.2” image sensor, SONY IMX249, and LED sources integrated with each camera. The selected camera resolution was 1920 × 1200 (2.3 megapixels), with a pixel size of 5.86 × 5.86 μm. Other specifications include an available field of view (FOV) of 150 × 200 mm and a detectable strain measurement range from 0.005% to 2000%. The VIC-EDU^®^ 8 System was placed 54.5 cm away from the specimen. A three-dimensional (3D) configuration was used to compensate for the effects of possible out-of-plane displacement of the specimens during testing. In addition, for better illumination of the specimen surface, two LED lights were added to the system. The VIC-Snap^®^ acquisition software included in the system was used to monitor the FOV of the cameras and verify the proper placement of the specimens. Likewise, the position of the LED sources was changed to verify and guarantee optimal lighting. The exposure time was about 50 milliseconds (ms) for all specimens tested. The implementation of the DIC method was divided into three stages, following this order: Specimen preparation by applying a specific speckle pattern, image acquisition, and post-processing.

The speckled pattern applied on the surface of specimens largely influences the correct resolution of the correlation problem and, consequently, also the performance of the DIC. Due to this, the surfaces of the specimens must exhibit a random pattern recognizable by the system (white/black ratio of 50:50) that ensures: high contrast with the background, isotropy, and minimal roughness enabling good tracking of the specimens. During preparation, a flat white base coat was applied evenly over the specimen surfaces using white base ink. A random pattern was then created using a dot patterning tool (toothbrush) embedded with black water-based ink and allowed to dry for approximately 10 min, the selected speckle size was around 0.1 mm. The calibration and image acquisition phases were performed using the VIC-Snap^®^ software. The DIC system was calibrated using a grid plate of dots with known spacing. In this way, the cameras of the system will correlate a physical distance with a pixel distance and determine their spatial location. The DIC system was calibrated by taking 30 images of the standard calibration plate with different positions and orientations. During the test, both cameras were set to capture an image every 2 s. After calibration, the image acquisition time was fixed considering the deformation time of the specimen, approximately 204 s. As a result, a total of 100 images were acquired for each specimen. Finally, the additional images after the rupture of the specimen were deleted before examining the results, obtaining an average of 30–40 images for each specimen. Regarding, the post-processing of the images stored in the VIC-3D^®^ software, an area of interest was defined for each specimen corresponding to its contour. Then, this area was divided into subsets of size large to contain enough information on the specimen. Based on the small dimension of the strain, the size of the subsets (9) was selected. Another determining parameter was the step size, which controls the number of pixels that separate each data point, this was set to 1 (minimum) to have more data points despite the longer analysis time. Then, if the points captured were very few or the projection error was high, these parameters were increased until the results were satisfactory. The process was completed by defining a starting point within the area of interest (AOI) from which the correlation process began, this was positioned at the bottom of the specimen. The main steps of the process are included in Figure 9. At the end of the post-processing phase, VIC-3D^®^ software tools were used to compute and display the strain variables in the area of interest on the image as follows. Normal strains: (e_yy_); the local longitudinal strain, Y-direction (following the direction of force application); (e_xx_); the local transverse strain, X-direction (positive values indicating stress, negative compression); and (e_eq_); the Von Mises strain, based on the distortion energy of a structure.

Since the software allowed us to calculate the strain between any two points within the area of interest covered by the speckle pattern, longitudinal and transversal virtual extensometers were placed on the images to investigate the deformation of the specimen. The Poisson’s ratio and local longitudinal strains of our interest were calculated as follows. Two extensometers were positioned along the longitudinal axis, *y*-axis (E_0_ and E_1_), and only one was placed transversal, *x*-axis (E_2_) to them. E_0_ was placed only along the narrow section ($L_{0}=33$ mm), E_1_ covered only one of the distal parts (L=16 mm) and E_2_ was placed perpendicularly, in the center of the narrow section. The Poisson’s ratio was computed through the relationship between axial and transverse deformations using the data collected by E_0_ and E_2_ virtual extensometers dividing the maximum values of strain detected at the same time point, respectively.

### 2.6. Statistical Analysis

All the mechanical parameters evaluated are presented as means ± standard deviations. Other parameters were computed to evaluate the stress–strain curve: Pearson’s coefficient (R2), to indicate the linearity, specifically the elastic region, and coefficient of variation (CV%), to assess the variability of the measurements.

## 3. Results

### 3.1. Finite Element Analysis, Evaluation of Pattern Position Inside the Specimens, and the Stress Distribution

Figure 10 shows the Von Mises stress distribution for the two cylindrical linear fiber patterns analyzed. There are no significant differences in terms of numerical results regarding the maximum stress values for configurations I and II.

### 3.2. Mechanical Characterization of Base PolyJet Materials Agilus30™—TissueMatrix™

The stress–strain curves obtained after the tests are shown in Figure 11. The calculated values of the mechanical properties by considering the mean ± standard deviation (std) values as well as Pearson’s coefficient (R^2^) and coefficient of variation (CV%) of the base PolyJet materials are summarized in Table A1 (Appendix A). Pearson’s coefficient values displayed resulted close to 1 for all the specimens (0.97–1). In the same way, the coefficient of variation calculated for the mechanical parameters does not show significant variations (1–9.26%).

### 3.3. Mechanical Results of Soft Tissue-Mimicking Polymers

The most significant stress–strain curves obtained after the tests are shown in Figure 12. The mechanical properties (mean ± std), as well as R2 and CV% of the soft tissue-mimicking polymers, are summarized in Table A2 (Appendix A). Pearson’s coefficient values displayed resulted to be close to 1 for all the specimens (0.96–0.99). In the same way, the coefficient of variation calculated for the mechanical parameters does not show significant variation (0.1–13.94%).

### 3.4. Digital Image Correlation Results of Soft Tissue-Mimicking Polymers

The advanced DIC non-contact instrumentation was used as a precise method to track and analyze in detail the volumetric deformation of patterned specimens during mechanical tensile testing. The set of representative images of each type of proposed pattern (A to S1) was stored and analyzed during the image acquisition and post-processing phases. The average values computed for the longitudinal, normal, and Von Mises strain fields (e_xx_, e_yy_, and e_eq_) along with Poisson’s ratio of the most significant specimens were reported in Table A3 (Appendix A), in addition to the values recorded by the virtual extensometers positioned in the regions of interest, that were used to calculate the Poisson’s ratio and for the comparative analysis of the values found in the strain fields. Moreover, the coefficient of variation (CV%) was reported to assess the variability of the measurements. The DIC strain fields shown in Figure 13 and Figure 14 taken in that exact frame are representative of one of the three specimens tested in a condition of maximum apparent elastic load before specimen rupture.

## 4. Discussion

### 4.1. Finite Element Analysis, Evaluation of Pattern Position inside the Specimens, the Stress Distribution Results

The simulation aimed to determine the appropriate position of the pattern before 3D printing the test pieces. An assessment of the influence of internal pattern placement (using a uniform diameter ø fiber pattern) within the entire test piece through both configurations is presented in Figure 10. In general, both test piece in-silico models exhibited, as expected, the highest stress values distributed along the test section. The numerical results in both configurations were similar, however, their location was different. The higher-stress values evidenced in the configuration I (0.89–0.96 MPa) were located in the change of CSA between the narrow section and the grips attachment. In this arrangement, as shown in Figure 3A, the fibers placed in the region of curvature of the test piece were not the same length as those placed along the narrow section, as well as these were not constrained by one of their ends. Then, in the shorter fibers that were subjected to uniaxial tension, shear stress was generated in correspondence with the joint radius, which caused a deficiency in this area of the specimen. On the other hand, configuration II, whose fibers arrangement was placed exclusively along the narrow section across the specimen as shown in Figure 3B, exhibited the higher-stress values (0.87–0.94 MPa) in the central region of the test piece as is expected during the mechanical tensile test. As known, mechanical test results could vary depending on the test piece geometry design and manufacturing process. Consequently, seeking repeatable comparative results and parameters we selected configuration II for multi-material 3D printing of the patterned specimens. This arrangement showed the Von Mises Stresses distribution proportional to the uni-axial applied load during the tensile test, evidencing the highest values in the central region (CSA is reduced as the strain increases) following the deformation of the test piece until its failure. These results were consistent with guidelines and recommendations specified in the standard preventing thus, early specimen failures during tensile testing in regions where there are CSA changes.

### 4.2. Mechanical Tensile Results Analysis

The results obtained from the mechanical tensile test are presented as follows. Figure 15 shows the methodology selected for the analysis. The mechanical tensile response of the base PolyJet materials, as well as the proposed soft tissue-mimicking polymers, was determined and the results compared.

### 4.3. Stress–Strain Curves from Uni-Axial Tensile Test

From the experimental data, the characteristic stress–strain graphs of the base PolyJet materials, and the most significant soft tissue-mimicking polymers of each pattern were plotted in Figure 11 and Figure 12. In general, all materials tested followed a representative ‘J’ stress–strain curve, characterized by a highly proportional linear elastic region, a non-well-defined plastic flow region, and a distinctive maximum tensile strength point reached after being subjected to the highest stresses. Mostly, the mechanical tensile response was guided by different tensile strengths depending on material or pattern, and large deformations all >100%. Moreover, the linearity of the elastic region that was evaluated using Pearson coefficients, varied from 0.97 to 0.99 for all tested materials. Values of R^2^ highlighted a highly linear correlation, which meant that data adjustment followed a suitable Hooke approximation in the proportional zone. Despite the linearity of the elastic region, in general, rubber-like PolyJet materials show the overall non-linear behavior and dependence on mechanical properties of the applied strain rate (viscoelastic) and build orientation as reported in several studies [26,41,72,73]. Finally, we found highly repeatable results across our entire mechanical tests performed driven by small standard deviations and CV% values ≤ 30%. The base PolyJet materials exhibited standard deviations which varied from 0.002 to 0.16, and CV% between 1.6–9%. Likewise, analyzing the soft tissue-mimicking polymers, these exhibited standard deviations which varied from 0.002 to 0.18 and CV% between 0.4–15.1%.

### 4.4. Mechanical Tensile Results of Base PolyJet Materials

To have a better understanding of the mechanical properties and behavior of proposed patterned specimens, a comparison of base PolyJet materials at different Shore A hardness was made. The materials showed a typical soft rubber-like behavior. As expected, polymers with a higher Shore A hardness exhibited lower strains with higher strength at break and elastic modulus values. In particular, the U1 specimens showed intermediate mechanical values with a strain at break corresponding to 170%, strength at break of 1.6 MPa, and modulus of elasticity of 1.72 MPa. On the other hand, the materials T1 and V1 specimens showed results close to those reported in the datasheet and the literature with deformations of 140–200%, strengths at break between 1.4–2.3 MPa, and modulus of elasticity equal to 1.1–2.2 MPa, respectively. Finally, W1 specimens showed the same intermediate strain at break of 170%, with the lowest values of strengths at break and elastic modulus as reported in Table A1 (Appendix A).

### 4.5. Mechanical Tensile Results of Tendon-Like Patterned Specimens

Analyzing the results summarized in Table A2 (Appendix A), the best variable combinations of the concentric fibers-matrix pattern were exhibited by Specimens A and C, in particular, Specimens C presented a 7% and 18% increase in their tensile strength at break and modulus of elasticity compared to the base material Agilus30™ A50. Despite that, simultaneously, these showed a 20% decrease of maximum strain at break, limiting its extensibility as failure stress increases. In this pattern type, the impact of the bio-inspired approach mimicking the hierarchical structure of the endotenon-fascicle-fiber did not obtain improvements in the mechanical performance. Indeed, the effect of adjusting the hardness value of the specimen matrix did not influence the tensile mechanical response of this group of specimens. Analyzing the results in detail, there was no combination of variables of this pattern that followed a defined trend or presented balanced mechanical properties based on the combination of the different hardness values between its layers as expected. In specimens A and E, as well as in C, the mechanical parameters were higher than the base material Agilus30™ A50 material but lower than the A60. On the other hand, the lowest values were presented by specimens D and F in which their mechanical performance decreased compared to the Agilus30™ A50.

### 4.6. Mechanical Tensile Results of Tendon-Mimic Patterned Specimens

In the second group of specimens, based on a Tendon-Mimic (TM) pattern, the mechanical tensile response followed a remarkable increase in its mechanical properties, these were better than the Agilus30™ A70 base material, the hardest one. The influential factor in the mechanical performance was the percentage ratio between non-uniform shaped fiber and matrix. The results were presented as follows. The specimens that showed the best mechanical performance were those belonging to the S-X subset, whose fiber-matrix ratio was 60–40%, respectively. In particular, Specimens V showed the best mechanical results, with a decrease of only 4% in its tensile strength at break, but with a significant increase of 35% in their maximum strain at the break while maintaining the same modulus of elasticity as regard the Agilus30™ A70 base material. In this type of pattern, the bio-inspired approach that mimicked the structure of the real anatomy of an animal tendon enabled us to determine the direct influence of a specific structural arrangement embedded in a soft-flexible matrix. Throughout the entire group of specimens in their three variant percentages of fiber-matrix, it was clear that the best mechanical results were influenced by a combination A70 fiber durometer and an A50 matrix. It was also found that the best mechanical results were obtained in the combinations with hard fibers and soft matrix, instead of variants in which the percentage of the area occupied by these and the durometer were higher. Initially, it could be thought that a higher percentage of the area occupied by the fibers or the matrix with higher hardness leads to better mechanical results, however, this pattern gives us a different interpretation regarding its tensile mechanical response.

### 4.7. Mechanical Tensile Results of Bamboo-Like Patterned Specimens

The third group of specimens characterized by a Bamboo-Like (BL) fibers-concentric pattern similar to the TL pattern, but with only one concentric fiber instead of multiple fibers, showed a mechanical tensile response followed by an increase in its mechanical properties. The BL pattern exhibited mechanical behavior widely better than the Agilus30™ A60 base material. Specimens Y and A1 showed the best results with similar mechanical values, but Specimens A1 showed higher values with a significant increase of 31% in their tensile strength at break but maintaining the same maximum strain at break and slightly increasing the modulus of elasticity by 21% compared to the Agilus30™ A60 base material. In this type of arrangement, the best results were influenced by combinations in which the inner and outer layers have been assigned the highest durometers. However, combination A1 together with combination D1 showed the lowest values of strain at break, this was mainly due to the highest durometers of their matrixes that limited its extensibility. In contrast, the combination C1 exhibited the best deformation, but with low values of tensile strength at break and modulus of elasticity.

### 4.8. Mechanical Tensile Results of Helix-Bamboo Patterned Specimens

The last group of specimens, based on a Helix-Bamboo (HB) double helix fibers-concentric pattern at different pitches according to the layer, but with only one concentric fiber, showed a mechanical tensile response followed by an increase in its mechanical properties. The best combinations were exhibited by Specimens J1 and O1. In particular, Specimens J1 exhibited mechanical behavior superior to the Agilus30™ A60 base material, evidencing the best tensile strength at break of all patterned specimens, with a 34% increase, 5% increase in their maximum strain at break, and 20% increase of the modulus of elasticity. Analyzing this type of pattern in detail, we also found that the subset E1 to H1 showed the highest strains at the break of the test, all of them shared a distinctive characteristic, their softest fibers, and matrix. Finally, the influence of the helix pattern as well as transitioning from soft-intermediate-hard durometers between layers did not have the desired effect in improving the tensile mechanical response while maintaining large strains.

### 4.9. Digital Image Correlation of the Soft Tissue-Mimicking Specimens

As expected, the transverse strain field e_xx_ highest values summarized in Table A3 (Appendix A) and displayed in Figure 13 and Figure 14 evidenced the effects of the contraction experienced by the specimens mainly along the narrow section as a consequence of the applied longitudinal load. The average values found for the most significant specimens (C, V, A1, J1) decreased in the cross-sectional area between −0.26 and −0.27. Likewise, the data recorded by the virtual extensometer E_2_ placed specifically in the central region of the specimens were close to those found in e_xx_. Specimens V experienced the greatest transverse contraction during the mechanical tensile test, followed by J1, and A1 while C was the one that exhibited the least transverse deformation. Indeed, the results found for the Poisson’s ratio confirm the decrease of the specimen gauge. of the soft tissue-mimicking polymers in the transverse direction, those evidenced the lowest values for specimens V and J1 and the highest for specimens C and A1. The tendency was analogous for the strain fields e_yy_ and e_eq_, the highest values found were located along the narrow section and in the central region of the specimen where the largest strains were experienced. Those varied between 1.39 and 1.53 for e_yy_ and 0.99 and 1.13 for e_eq_, respectively. On the other hand, it was comparing the values for virtual longitudinal extensometer E_0_ in combination with the images acquired by the DIC system, (specifically, those representatives of the specimens at the beginning of the test and just before failure) and strain values obtained applying the ASTM D638. It was evidenced that the strain measured with both methods was different. The strain recorded by the digital extensometer E_0_ was smaller than the strain values computed following the standard. This was mainly because the tensile machine recorded the strain values experienced by the specimen along its entire test section (L=65 mm) considering the deformation of the distal parts of the specimen as well, regardless that the calculated strain was divided by the dimension of the narrow section (L=33 mm). This was verified, following the values recorded by the E_1_ virtual extensometer, where for the most significant specimens the deformation of the distal part was not negligible and varied between 0.68 and 0.7. Therefore, despite, the extensometer E_0_ can vary considerably concerning where it is positioned, it records deformation values more accurately. For the most significant specimens (C, V, A1, J1), specimens V experienced the greatest longitudinal deformation and consequently, the greatest transverse compression followed by Specimens J1, A1, and lastly C.

### 4.10. The Most Significant Soft Tissue-Mimicking Specimens, Comparative Analysis

The experimental results of the best soft tissue-mimicking specimens and the Agilus30™ base PolyJet material at different Shore A hardness levels were summarized and compared as shown in Figure 16. The specimens V (Tendon-Mimic) showed best balanced mechanical properties reflected in terms of higher values of modulus of elasticity (MPa), maximum strain at break (mm/mm), and values of tensile strength at break (MPa) along with those evidenced by specimens J1 (Helix-Bamboo). Likewise, comparing them with the Agilus30™ base PolyJet specimens. These presented slightly higher values of modulus of elasticity (MPa) and near values of tensile strength at break (MPa) to the hardest combination, Agilus30™ A70 base material. Evaluating its maximum strain at break (mm/mm), although it is not close to the values reported by the Agilus30™ A50 base material, the softest one, it exceeds the Agilus30™ A60 and A70 combinations.

As known, connective soft tissues, in particular tendons and ligaments are characterized by high values of modulus of elasticity (between 0.2 to 1.5 GPa) exhibiting non-linear anisotropic, viscoelastic (mechanical properties are rate and history-dependent), and adaptive behavior which is quite complex to reproduce. The unique structures act and transmit musculoskeletal forces by providing stability and a wide ROM changing shape as a result of their ability to withstand large strains (18% to 30%), and tensile strength at break (4.4 up to 660 MPa). In addition to this, although both structures exhibit similar mechanical behavior, despite their anatomical attachment sites, these mechanically behave differently based on the physiological features and functional requirements. For example, shorter soft tissues with a thicker CSA exhibit a higher modulus of elasticity and tensile strength at break. On the other hand, flat and long native structures reach higher strains before failure. Additional factors, such as orientation and positioning of structures, pathological conditions, age, and gender also affect their mechanical properties. Indeed, degenerative soft tissue disorders show a reduction in their mechanical properties, exhibiting a lower modulus of elasticity, strength, and strain compared to healthy controls [40,68,74,75]. In normal conditions, generally, when a progressive mechanical load is applied to a tendon or a ligament, the collagen bundles, which are randomized and crimped at rest, start to align with each other losing the initial state that represents a 2–5 strain percentage in their stress–strain curve (toe region). Continued tissue-loading results in increasing stiffness until a stage is reached where they exhibit linear behavior. The collagen bundles are straightened and contribute to sustaining the applied load providing a quite ideal elastic recovery, in terms of a linear relationship of stress and strain (elastic region). Then, collagen bundles continue to absorb energy progressively until complete the tensile failure. Their stress–strain curve is characterized by a non-well-defined plastic region that achieves the failure point. It is well recognized that polymers are adequately suited to mimic some complex aspects of the organic systems, since these exhibit similar mechanical behavior by reproducing some mechanical properties, and the typical ‘J’ shape of the soft tissue stress–strain curve [55,58,68,69]. Our experimental study was not intended to replace native tissues, but rather to take advantage of the customization of the polymeric materials available in the Stratasys J750™ DAP to integrate different bio-inspired structural and hierarchical patterns into a single construct. The described approach provided an improved mechanical response, superior to single base PolyJet materials, allowing to obtain a customized structure, that can mimic the mechanical behavior and ROM of tendons and ligaments without failing. To achieve an improved mechanical response, it was necessary to overcome the limitations presented by single-base PolyJet materials. For example, Agilus30™ A70 was the most strength material but at the same time the least deformable, in contrast, Agilus30™ A50 material was the most flexible but least strength, and Agilus30™ A60 does not present the mechanical balance required for the application. Therefore, these options cannot simulate the elasticity and stability of real tissues when they are subjected to mechanical stress.

On the other hand, there was the option of manufacturing a composite material using the J750™ DAP capabilities, combining a hard phase (a rigid plastic) and a soft phase (a rubber-like). However, this approach presents several disadvantages such as material transitions, compatibility, or hard phase fracture. The use of a composite material with these drawbacks would lead to premature failure of the construct by subjecting it to large deformations. overcoming these limitations, our approach was the most suitable option for manufacturing the structure, when comparing the soft tissue-mimicking specimens to native tendons and ligaments, this method suggests that Stratasys J750™ DAP and its rubber-like materials can create constructs that approach the targeted tensile mechanical response of these structures. Although quantitatively our mechanical values are not close to those reported for real tissues, the stress–strain curves reported may overlap. Except for the typical toe region, soft tissue-mimicking specimens behave similarly exhibiting a proportional elastic region, a non-well-defined plastic region, and strength and failure limits. Moreover, the mechanical properties reported in this study and those described in the literature for related soft tissue applications are similar to, for example, HydroThane™ (hydrophilic thermoplastic polyurethane), a material used to fabricate soft-tissue medical devices [76,77]. The described method could be adapted to the clinical and research needs of 3D printed soft tissue anatomical models. Finally, additional mechanical characterization tests, such as compression and fatigue, will increase the understanding of the behavior of the polymeric materials described in this study.

## 5. Conclusions

Following a bio-inspired approach, it was investigated the mechanical tensile response of proposed soft tissue-mimicking polymers with embedded hierarchical patterns using UT and DIC. Mechanical properties and representative curves with native tendon and ligament values and behaviors were compared. After data analysis, the next conclusions can be formulated:Results evidenced a difference in the mechanical properties of the specimens according to pattern type, proposed Shore A hardness combinations, and matrix-to-fiber ratio. The soft tissue-mimicking specimens V, J1, A1, and C were selected as the best ones for every type of pattern proposed. In particular, specimens V with a high Shore A value assigned to fibers with non-uniform morphology contained in a softer matrix allowed to obtain a comparable tensile strength at break and a higher total deformation of the specimen compared to the base material Agilus30™ A70. This configuration compensates for the reduced deformability/extensibility and/or reduced fatigue life of a possible single-material anatomical model using.The approach demonstrates the customization capacity and versatility of the J750™ DAP. This technology allowed to set different Shore A values to the various components of the pattern in combination with the design of bioinspired geometries, which led to superior mechanical behaviors compared to the base PolyJet materials with a fixed durometer. Our results followed a qualitative approach mimicking the mechanical response of tendon and ligament structures, these will be useful for the development of more accurate and realistic computational, and 3D printed soft tissue anatomical solutions mimicking something much closer to real tissues. Particularly, the major difference is in the initial portion of the stress–strain curve (toe region), this specific behavior unfortunately is not reproducible with the current approach; however, the scope of this research is focused on the global behavior at relatively large strains, thus not on the initial portion of the stress–strain curve.The J750™ DAP and its dedicated software provide the ability to create and fabricate high-fidelity anatomical and pathological models using sophisticated anatomical presets configured using material combinations and transitions that can vary in stiffness, flexibility, and density. Depending on the future application, the patterns explored in this study, in particular, the TM pattern could be included as an anatomical preset exploring the potential to fabricate 3D printed tendon and ligament structures with better mechanical characteristics than base PolyJet and composite materials.Thanks to its capabilities (high resolution and micrometric layer thickness), the J750™ DAP allowed for obtaining very precise repeatability of the mechanical tensile test (average std and CV% reported). At the same time, PolyJet technology enabled the creation of specimens with mechanical characteristics of tensile and performance customized that can fit specific research and clinical needs to the application to be achieved.ASTM standard test piece shapes used to determine the mechanical tensile properties of AM rubber-like materials may not be most appropriate for testing. As evidenced by the DIC results, the distal parts of the specimen presented significant deformations that are not considered for the determination of the mechanical properties according to the standard. Therefore, the development of new tensile specimen shapes or tensile test methods for 3D printed specimens that do not include variable CSA geometries may be required.Native tissues are complex and contain combinations of materials and transitions with one or more types of structural patterns that start from a constituent unit, often at the nanometric level, which is very challenging to mimic and manufacture. Knowledge of mechanical design parameters such as modulus of elasticity, tensile strength, maximum strain at break, and Poisson’s ratio is fundamental for characterizing and establishing new future similar approaches, including rubber-like PolyJet materials.

## Figures and Tables

**Figure 1 polymers-14-02639-f001:**
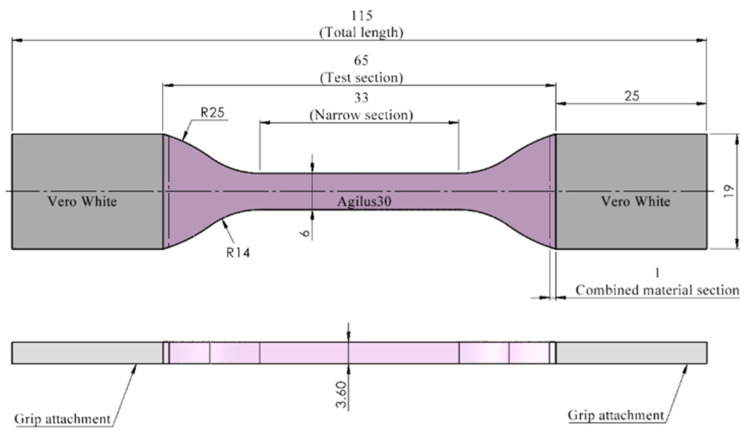
The dog-bone specimen Type IV shape and dimensions according to ASTM Standard D638. Total length 115 mm; test section 65 mm; narrow section 33 mm; grips attachments 26 mm × 19 mm; combined material section 1 mm; width; 6 mm; thickness 3.6 mm.

**Figure 2 polymers-14-02639-f002:**
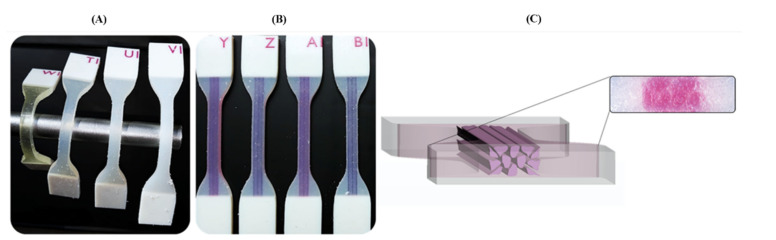
Three-dimensional printed dog-bone specimens of the different base PolyJet materials and the proposed biomimetic patterns. (**A**) Comparison of the different durometers’ levels of the Agilus30™ and TissueMatrix™ materials. (**B**) The specimens with the different embedded patterns are shown. The test piece combines the rigid (VeroWhite™) and flexible (Agilus30™) materials. (**C**) A cross-section of the Tendon-Mimic (TM) specimen is shown in a microscope view at 100× magnification.

**Figure 3 polymers-14-02639-f003:**
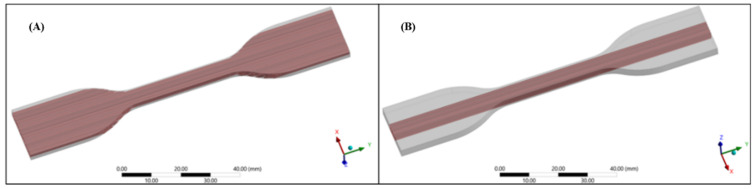
View of the dog-bone specimens used in Finite element analysis at Ansys structural environment. (**A**) Pattern configuration I fibers are uniformly distributed along the entire specimen’s volume (**B**) Pattern configuration II fibers are positioned only in the narrow cross-section of the specimen.

**Figure 4 polymers-14-02639-f004:**
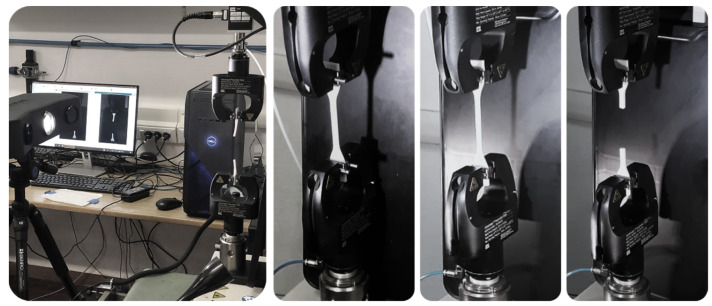
Experimental mechanical testing setup. The deformation of the dog-bone specimen in the initial elastic stage until failure is shown.

**Figure 5 polymers-14-02639-f005:**
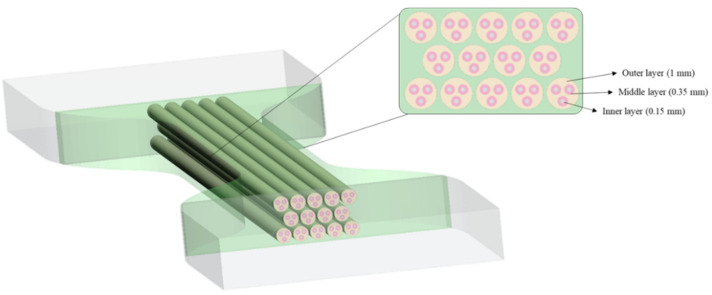
Tendon-Like (TL) pattern arrangement. In a cross-sectional view, the dimensions of each layer are reported.

**Figure 6 polymers-14-02639-f006:**
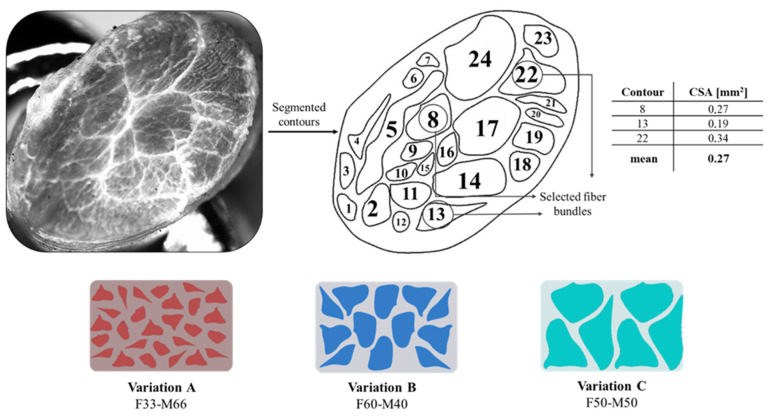
Tendon-Mimic (TM) pattern arrangement. The segmented contours of the equine tendon and cross-sectional areas are reported, as well as the percentage of area occupied by the fibers on the narrow cross-section of the specimen.

**Figure 7 polymers-14-02639-f007:**
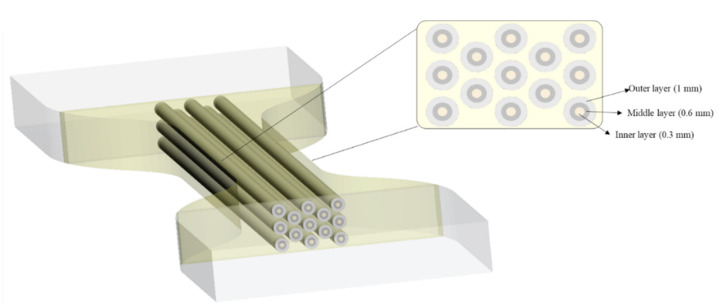
Bamboo-Like (BL) pattern arrangement. In a cross-sectional view, the dimensions of each layer are reported.

**Figure 8 polymers-14-02639-f008:**
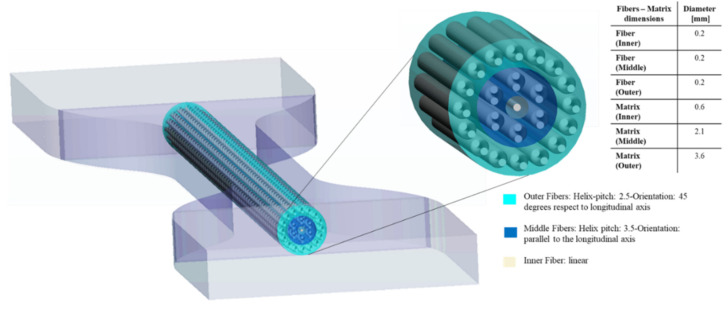
Helix-Bamboo (HB) pattern arrangement. In a cross-sectional view, the dimensions, helix-pitch, and orientation of each layer are reported.

**Figure 9 polymers-14-02639-f009:**
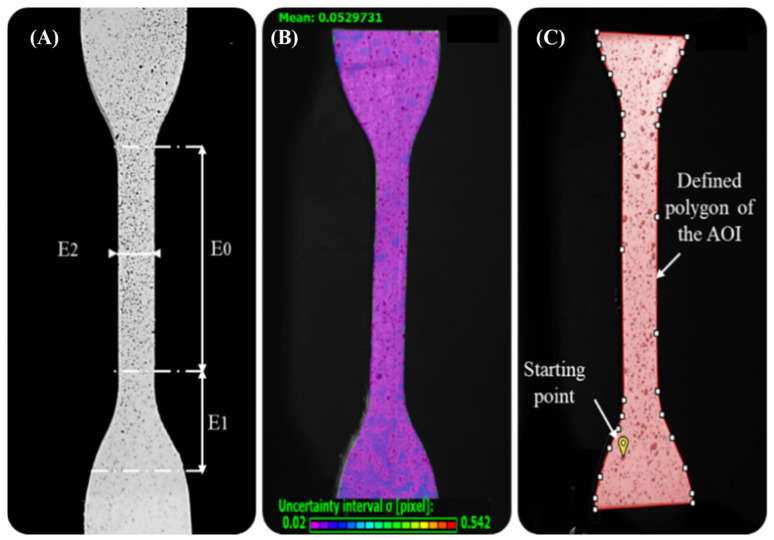
Implementation of the Digital Image Correlation (DIC), main stages. (**A**) Speckle pattern applied on the dog-bone specimen Type IV surface before the tensile test, the defined position of the virtual extensometers E_0_, E_1_, and E_2_ are along the XY axes in the regions of interest. (**B**) Image acquisition of the specimen, uncertainty range in the correlation of subsets in images; purple: maximum correlation-red: none. (**C**) Post-processing phase. Area of Interest (AOI) on which VIC-3D performs correlation analysis. defined by a polygon on the specimen surface image. Likewise, the starting point is evidenced placed in the area of the image that is subjected to the least amount of movement (grip attachment) during the tensile test.

**Figure 10 polymers-14-02639-f010:**
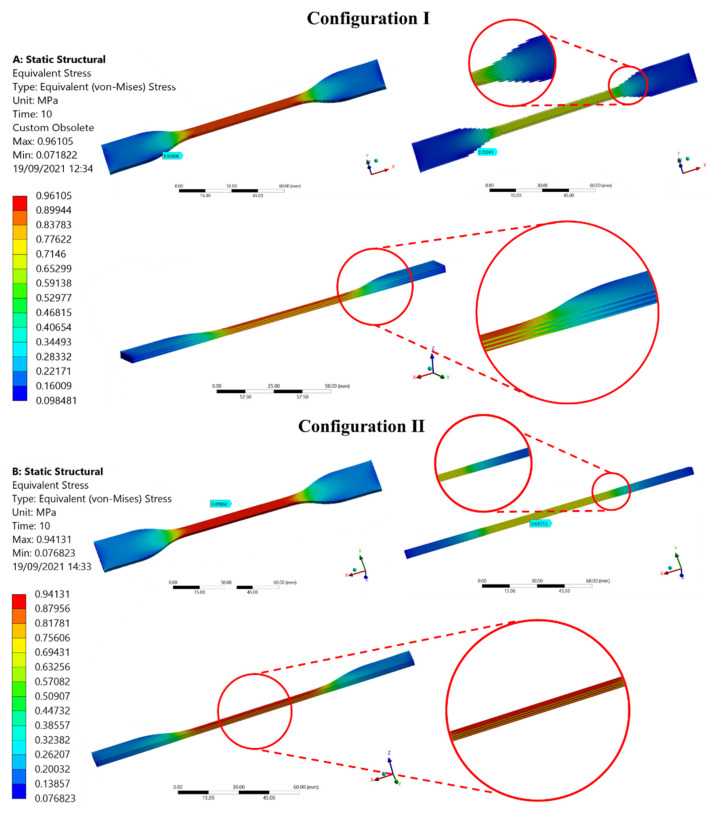
Ansys static structural environment. Results of Von Mises Stress distribution in configurations I and II.

**Figure 11 polymers-14-02639-f011:**
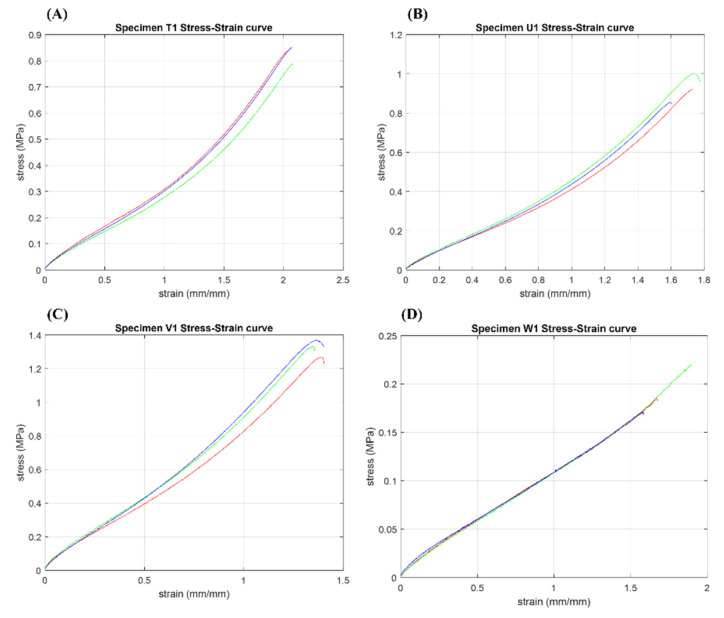
The stress–strain curves obtained after mechanical testing of base PolyJet materials. Specimens ID: (**A**) T1 = Agilus30™ base material-Shore hardness selected A50. (**B**) U1 = Agilus30™ base material-Shore hardness selected A60. (**C**) V1 = Agilus30™ base material-Shore hardness selected A70, and (**D**) W1 for TissueMatrix™.

**Figure 12 polymers-14-02639-f012:**
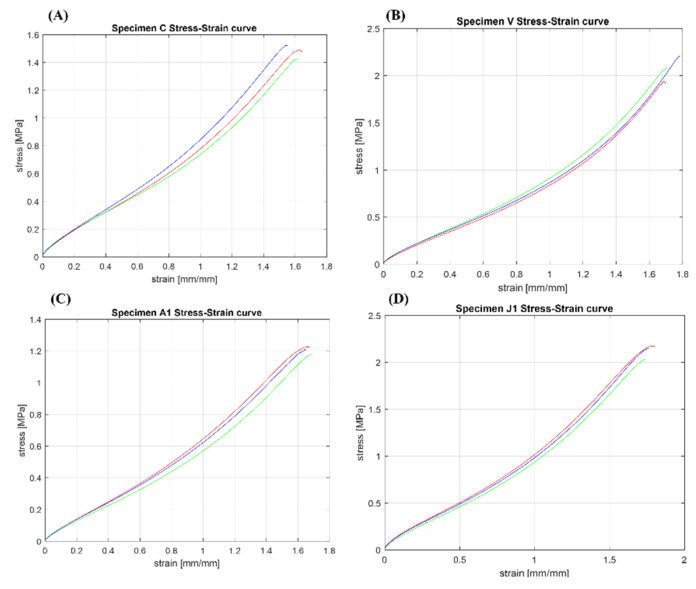
The most significant stress–strain curves obtained after mechanical testing of patterned specimens. (**A**) Tendon-Like (TL) pattern: C = Inner Layer: Shore hardness A70—Middle Layer: Shore hardness A60—Outer Layer: Shore hardness A70, and Matrix: Shore hardness A60. (**B**) Tendon-Mimic (TM) pattern: V = Variant %: Fiber 60%—Matrix 40%, Fiber: Shore hardness A70, and Matrix: Shore hardness A50. (**C**) Bamboo-Like (BL) pattern: A1 = Inner Layer: Shore hardness A70—Middle Layer: Shore hardness A60—Outer Layer: Shore hardness A70, and Matrix: Shore hardness A60. (**D**) Helix-bamboo (HB) pattern: J1 = Inner Layer: Shore hardness A50—Middle Layer: Shore hardness A60—Outer Layer: Shore hardness A70, Fiber and Matrix: Shore hardness A60.

**Figure 13 polymers-14-02639-f013:**
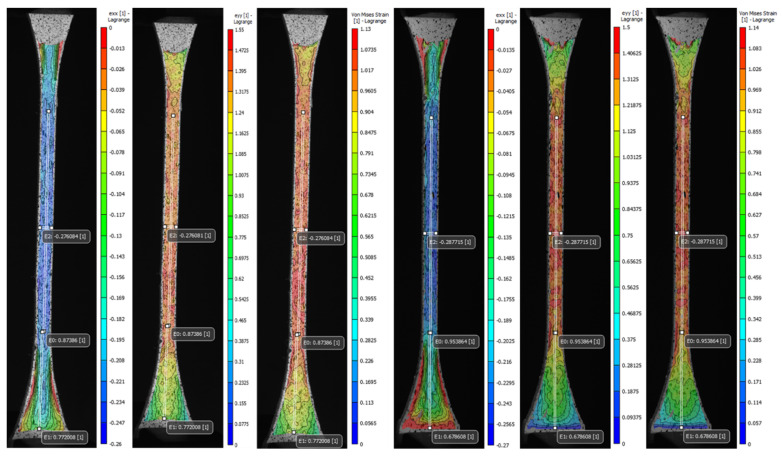
Digital image correlation (DIC) strain fields of the most significant specimens: V Tendon-Mimic (TM)—J1 Helix-Bamboo (HB) patterns. Longitudinal strain (e_yy_), local transverse strain (e_xx_), and Von Mises equivalent strain (e_eq_). The virtual longitudinal extensometers E_0_ and E_1_ are positioned at $L_{0}=33$ mm (narrow section) and L=16 mm (distal). Transverse extensometer E_2_ is positioned at L=6 mm (central).

**Figure 14 polymers-14-02639-f014:**
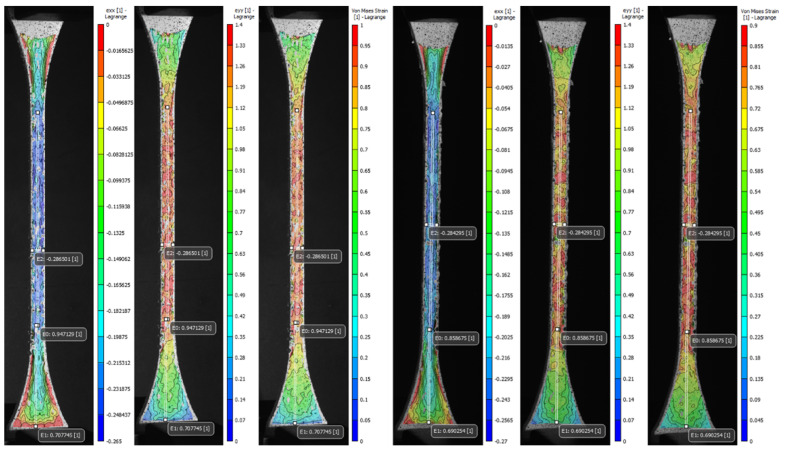
Digital image correlation (DIC) strain fields of the most significant specimens. A1 Bamboo-Like (BL)—C Tendon-Like (TL). Longitudinal strain (e_yy_), local transverse strain (e_xx_), and Von Mises equivalent strain (e_eq_). The virtual longitudinal extensometers E_0_ and E_1_ are positioned at $L_{0}=33$ mm (narrow section) and L=16 mm (distal). Transverse extensometer E_2_ is positioned at L=6 mm (central).

**Figure 15 polymers-14-02639-f015:**
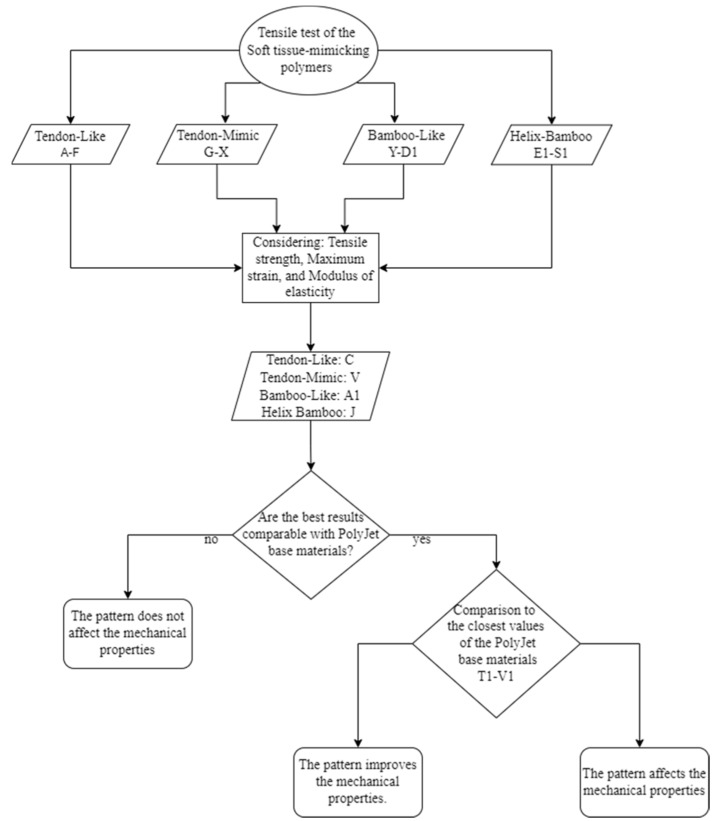
The workflow used in the experimental study for the analysis of the mechanical tensile results. Pattern Types: Tendon-Like (TL). Tendon-Mimic (TM). Bamboo-Like (BL). and Helix-Bamboo (HB). Specimen ID: C durometers; Inner layer A50-Middle layer A60-Outer layer A70-Matrix A60. Specimen ID: V variant percentage of the area occupied; Fiber 60%-Matrix 40%-durometers; Fibers A70-Matrix A50. Specimen ID: A1 durometers; Inner layer A70-Middle layer A60-Outer layer A70-Matrix A60. Specimen ID: J1 durometers; Fiber and matrix A60-Inner layer A50-Middle layer A60-Outer layer A70-Matrix A60.

**Figure 16 polymers-14-02639-f016:**
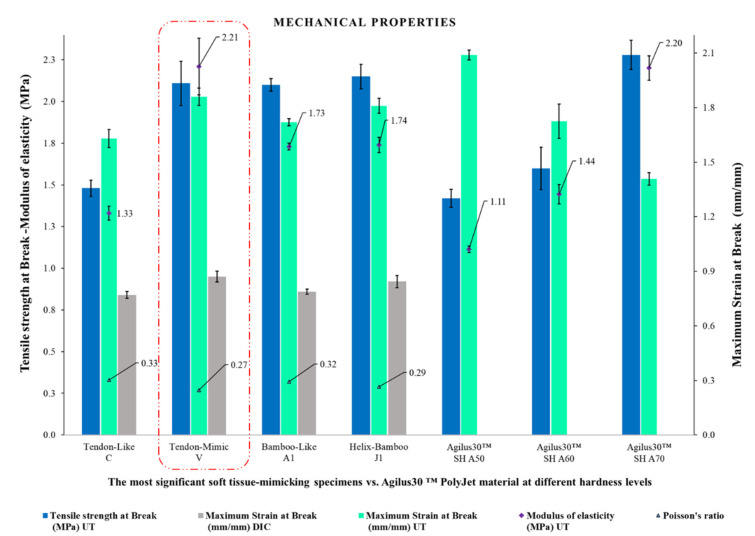
Comparison between the mechanical parameters (modulus of elasticity, tensile strength at break, maximum strain at break measured by UT and DIC, and Poisson’s ratio) of the specimens C, V, A1, and J1, and base PolyJet material Agilus30™ at different Shore A hardness levels, DIC measurements were not performed for base PolyJet polymers.

**Table 1 polymers-14-02639-t001:** Shore A hardness values were assigned for each layer of the Tendon-Like (TL) pattern. The range considered was from 50 (medium-soft) to 70 (hard-soft).

Pattern Type	Specimen ID	Inner Layer	Middle Layer	Outer Layer	Matrix
**Tendon-Like (TL)**	A	50	60	70	60
	B	70	60	50	60
	C	70	60	70	60
	D	50	60	50	60
	E	70	50	70	50
	F	50	70	50	70

**Table 2 polymers-14-02639-t002:** Shore A hardness value was assigned for each layer of the Tendon-Mimic (TM) pattern. The range considered was from 50 (medium-soft) to 70 (hard-soft).

Pattern Type	Variant %	Specimen ID	Fiber	Matrix
**Tendon-Mimic (TM)**	**Fiber 33%** **Matrix 66%**	G	50	60
		H	60	50
		I	50	70
		J	70	50
		K	60	70
		L	70	60
	**Fiber 50%** **Matrix 50%**	M	50	60
		N	60	50
		O	50	70
		P	70	50
		Q	60	70
		R	70	60
	**Fiber 60%** **Matrix 40%**	S	50	60
		T	60	50
		U	50	70
		V	70	50
		W	60	70
		X	70	60

**Table 3 polymers-14-02639-t003:** Shore A hardness values were assigned for each layer of the Bamboo-Like (BL) pattern. The range considered was from 50 (medium-soft) to 70 (hard-soft).

Pattern Type	Specimen ID	Inner Layer	Middle Layer	Outer Layer	Matrix
**Bamboo-Like** **(BL)**	Y	50	60	70	60
	Z	70	60	50	60
	A1	70	60	70	60
	B1	50	60	50	60
	C1	70	50	70	50
	D1	50	70	50	70

**Table 4 polymers-14-02639-t004:** Shore A hardness values were assigned for each layer of the Helix-Bamboo (HB) pattern. The range considered was from 50 (medium-soft) to 70 (hard-soft).

Pattern Type	Specimen ID	Fiber and Matrix	Inner Layer	Middle Layer	Outer Layer
**Helix-Bamboo** **(HB)**	E1	50	60	50	60
	F1		60	70	60
	G1		70	50	70
	H1		70	60	70
	J1	60	50	60	70
	K1		70	60	50
	L1		50	60	50
	M1		70	50	70
	N1		50	70	50
	O1		70	60	70
	P1	70	50	60	50
	Q1		50	70	50
	R1		60	50	60
	S1		60	70	60

## Data Availability

The data associated with this article used to support the findings of this study are available from the corresponding author on reasonable request.

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
