# Peer review of "Design and Mechanical Characterization Using Digital Image Correlation of Soft Tissue-Mimicking Polymers"

_polymers, 2022, doi:10.3390/polym14132639_

Round 1
Reviewer 1 Report
The authors reported a fully-described study about a rationale design and implementation of polymeric hierarchical structures bioinspired by tendons with different fibres organisations.
In my opinion, the work is suitable for publication, I report in the following some minor comments/suggestions:
1) the manuscript is quite long. There are a lot of tables and descriptions, but probably not all of them have the same importance in order to follow the workflow and discussion. I suggest putting the “extra” material in a “supplementary materials” section.
2) Similar comment for e.g., the introduction: it is too long, I would suggest reducing it, summarising the main concepts.
3) With reference to the second point in the conclusions and the graphs in Fig. 11 and 12, a different initial part of the stress-strain curves with respect to native tendons is observed (especially when referring to the "toe" region). This typical region is mainly due to the uncrimping of the collagen fibres, which I suppose is not possible to reproduce within this approach. I am wondering how this different behaviour could affect the final performance of an artificial tendon or a ligament. Could the authors provide a comment on this?
Reviewer 2 Report
This study presents an important subject of biomedical approach. Some flaws were detected and described below.
Abstract: the abstract is poorly structured. It presents methodological and results absence. Consider removing the introduction and reducing the objectives.
The introduction of this study is complete and scientifically based, however, it presents serious problems of textual coherence. Paragraphs that are too long and with inconsistent sentences.
[151-152] “Lastly, the experimental results are presented and discussed, followed by conclusions regarding the application of the proposed approach.” This is an unnecessary information.
Materials and methods: consider improving the statistical analysis description. Due to the large number of methods performed and results, it becomes confusing to understand the statistical analysis performed. A specific topic at the end of materials and methods is advised, as well as a presentation of the results in the same sequence of presentation of the materials and methods.
Fig 16. Consider changing the position of this figure to results topic. Statistical analysis was performed? If so, insert statistical representations in the figure.
Reviewer 3 Report
In this manuscript, the authors report some mechanical characterization including modulus of elasticity, tensile strength at break, maximum strain at break and Poisson's ratio for 3D printed multi-material dog-bone specimens, which are designed with different combinations of three Shore-A values (Agilus30™ Shore hardness A50, A60, A70) and four bio-inspired patterns (Tendon-Like, Tendon-Mimic, Bamboo-Like and Helix-Bamboo). The results they presented show a complex landscape for the mechanical properties of these specimens, as the authors indicated, the difference in these mechanical properties are depending on the pattern type, proposed Shore A hardness combinations, and matrix-to-fiber ratio. Nevertheless, the authors have identified specimens Tendon-Like C, Tendon-Mimic V, Bamboo-Like A1, and Helix-Bamboo J1 as the best ones for every type of pattern proposed, based on their analysis on data obtained from the combination of uni-axial tensile tests and digital image correlation. These results are meaningful guides for further optimization in the design of biomimetic hierarchical patterns for various 3D printed tissues.
However, there are some shortcomings to be revised if it to be published:
(1) Some writings seem to be too repeating and redundant. Suggest that some data and their reports involving in the results 3.3 and 3.4 be refined, e.g., Fig.11 and Fig.12, may be modified from their respective four sub-figures to one united figure, so as to ease reader more visual comparison for the different four specimens; While the section 3.2 may be removed to proper place in ‘2. Materials and Methods’.
(2) Some full stops might be printed wrong places, such as in line 21, 622, pls have a check. The sentence at p46-47 seems to be incomplete. Maybe, the interpunction before it uses wrong.
In Fig.2A, this is not difficult to show all the word labels on the respective specimens but now is lacking W1. And why use different color to present the microscope view in Fig.2C? It feels not matched.
Reference 38 prints superfluous words ‘Corresponding author’.
